# Discrimination and Symptoms of Post-Traumatic Stress Among Black Transgender Women in the United States: The Moderating Effect of Sleep

**DOI:** 10.3390/healthcare14020137

**Published:** 2026-01-06

**Authors:** Monique S. Balthazar, Lindsay Master, Daniel Jackson Smith, Athena Sherman

**Affiliations:** 1Ross and Carol Nese College of Nursing, Pennsylvania State University, University Park, PA 16802, USA; 2Department of Biobehavioral Health, College of Health and Human Development, Pennsylvania State University, University Park, PA 16802, USA; lmaster@psu.edu; 3School of Nursing, University at Buffalo, Buffalo, NY 14260, USA; dsmith55@buffalo.edu; 4Nell Hodgson Woodruff School of Nursing, Emory University, Atlanta, GA 30322, USA; adfsherman@emory.edu

**Keywords:** mental health, sleep, transgender, discrimination, intersectionality, PTSD, cultural validity, structural determinants, health disparities

## Abstract

**Background**: Black transgender women experience high rates of intersectional discrimination contributing to post-traumatic stress disorder (PTSD) symptoms. While sleep typically buffers psychological distress among general populations, these relationships remain underexplored among Black transgender women, and existing protective sleep literature derives primarily from non-Hispanic White, cisgender, socioeconomically advantaged populations. **Methods**: This exploratory secondary cross-sectional analysis of 155 Black transgender women (aged 18+) examined whether sleep quality (Pittsburgh Sleep Quality Index) moderates associations between discrimination (Intersectional Discrimination Index) and PTSD symptoms (PTSD Symptom Checklist-DSM-5) using moderated multiple linear regression models, controlling for age (n = 139–149). **Results**: Contrary to expectations, better sleep quality strengthened associations between day-to-day (*p* = 0.0126) and major discrimination (*p* = 0.0235) and the PTSD symptom severity. **Conclusions**: These exploratory findings reveal paradoxical sleep-distress relationships among Black transgender women that contradict patterns documented among general populations, highlighting critical limitations in applying existing psychological frameworks to multiple marginalized communities. Results underscore urgent needs for culturally validated assessment instruments and comprehensive measurement of structural determinants (housing stability, economic security, and neighborhood safety) before concluding psychology in populations experiencing intersectional oppressions.

## 1. Introduction

Transgender women (individuals who were assigned male at birth and identify as women) face pervasive discrimination, with 70% reporting discriminatory experiences within the prior 12 months [1]. For Black transgender women specifically, discrimination operates at the intersection of multiple marginalized identities, creating compounded risks that extend beyond single-axis experiences of either racism or transphobia alone [2,3]. The relationship between discrimination and mental health outcomes among transgender women is well-established, with PTSD prevalence ranging from 18% to 61%, approximately 2.5 times higher than in cisgender populations [4,5]. Sleep disturbances also represent a profound disparity, with 79.2% of transgender women reporting sleep problems [6].

Sleep quality typically functions as a protective factor against discriminatory stress among general populations studied to date (primarily cisgender, socioeconomically advantaged samples) [7,8,9,10,11]. This protective effect is theorized to operate through two primary mechanisms: the allostatic load framework, which describes how restorative sleep buffers the physiological and psychological impact of chronic stressors such as discrimination [12,13,14], and the Rapid-Eye-Movement (REM) sleep hypothesis of emotional-memory processing, which posits that REM sleep enables overnight processing of emotional experiences, reducing amygdala reactivity [15,16].

However, it is critical to recognize that Black transgender women’s experiences may fundamentally differ from those captured in the existing sleep literature. Chronic exposure to structural racism, housing instability, economic marginalization, violence, and systemic barriers to healthcare [17,18] creates conditions qualitatively distinct from those experienced by populations in which protective sleep effects have been documented. The unique stressors faced by Black transgender women, including transmisogyny, racialized violence, and compounded economic vulnerability, may fundamentally alter protective relationships observed in more privileged populations. Therefore, hypotheses derived from general population research must be tested rather than assumed for multiply marginalized communities.

This exploratory study examines how sleep quality moderates relationships between experiences of different types of discrimination (anticipated, day-to-day, and major) and the severity of PTSD symptoms among Black transgender women. Given the absence of prior research on sleep as a moderator in this population, we approach this inquiry without directional hypotheses, remaining open to the possibility that conventional protective effects of sleep quality may not operate as anticipated within this population.

## 2. Methods

### 2.1. Study Design, Setting, and Participants

The current article reports on a secondary analysis of cross-sectional electronic survey data collected from 155 Black transgender women in the United States (US) from October 2021 to February 2024. The original study was a mixed-methods study which collected quantitative and qualitative data; however, this article only reports on the quantitative data. Eligibility criteria included (1) current gender identity must be a feminine gender such as woman, transgender woman, transfeminine; (2) sex assigned on their original birth certificate must be male or intersex (if intersex, then must self-identify with the term transgender); (3) identify their race as Black, or of multiple-races to include Black; (4) reside in the US; (5) be 18 years or older; and (6) be able to read and comprehend English. Exclusion criteria included the presence of a psychotic or bipolar disorder not currently controlled with medication, as assessed by clinician-administered mini international neuropsychiatric interview [19] modules C and K during screening. The Emory Institutional Review Board approved the original study, IRB#: STUDY00002141. The original research detailed methods are reported here [20].

### 2.2. Data Collection

Participants completed an electronic survey via REDCAP (https://project-redcap.org/) that lasted on average ~45 min. The survey collected data on several demographic variables, health indicators, and predictors of health (e.g., sleep, depression, PTSD, suicidality, quality of life, violence exposure, discrimination exposure, housing status, age, race, income, etc.). For this analysis, we used the following measures:

#### 2.2.1. Independent Measures: Intersectional Discrimination

The Intersectional Discrimination Index (InDI) [21] measures discrimination experience across three dimensions. Mean reported scores for experiences of anticipated discrimination (nine items, range 0–4; example item: “Because of who I am, people might try to attack me physically.”), experiences of day-to-day discrimination (nine items, range 0–3; example item: “Because of who you are, have you been told that you should think, act, or look more like others?”), and experiences of major discrimination (thirteen items, range 0–2; example item: “Because of who you are, has a health care provider ever refused you care?”) were calculated. Participants must have 80% of the items as non-missing from each subscale to be included in the analysis. Higher scores indicate more reported discrimination. In the present study, internal consistency was excellent for all subscales: InDI-Anticipated (Cronbach’s α = 0.92), InDI-Day-to-day (Cronbach’s α = 0.93), and InDI-Major (Cronbach’s α = 0.90). Each of the three scales was treated as a continuous variable.

#### 2.2.2. Moderators: Sleep Quality

Seven subjective sleep components (sleep quality, latency, duration, efficiency, disturbance, use of medication, and daytime dysfunction) and a global score were calculated using the Pittsburgh Sleep Quality Index [22]. Example items include “During the past month, how would you rate your overall sleep quality?” (rated very good, fairly good, fairly bad, or very bad) and “During the past month, how many hours of actual sleep did you get at night?” (converted to scaled scores). Each of the seven components was scored into four categories (0–3, where zero is typically associated with no sleep difficulty and three is typically associated with severe sleep difficulty). The global score ranges from 0–21, where higher scores indicate worse sleep quality. In the present study, internal consistency for the global PSQI total score was acceptable (Cronbach’s α = 0.70). Among individual components with at least two items, internal consistency varied: sleep latency (two items, α = 0.58), daytime dysfunction (two items, α = 0.60), and sleep disturbance (nine items, α = 0.83). All sleep moderator variables were treated as continuous variables.

#### 2.2.3. Dependent Measure: Post-Traumatic Stress

Post-traumatic stress was measured using the PTSD Symptom Checklist for DSM-V (PCL-5) [23,24], a twenty-item self-report assessing symptoms across four DSM-5 clusters: intrusion (example: “repeated, disturbing, and unwanted memories of the stressful experience?”), avoidance (example: “Avoiding memories, thoughts, or feelings related to the stressful experience?”), negative alterations in cognitions and mood (example: Trouble remembering important parts of the stressful experience?”), and alterations in arousal and reactivity (example: “Irritable behavior, angry outbursts, or acting aggressively?”). Each item is rated on a five-point scale (0 = not at all and 4 = extremely). Higher scores indicate higher symptoms of post-traumatic stress (possible range 0–80), diagnostic threshold of 31 points [23,24]. In the present study, internal consistency for the PCL-5 total score was excellent (Cronbach’s α = 0.96).

#### 2.2.4. Statistical Analysis

Separate moderated multiple linear regression models in SAS 9.4 were conducted to examine whether self-reported sleep moderates the relationship between discrimination and post-traumatic stress. Regression assumptions were checked before the analysis, including linearity, normality of residuals, homogeneity of variance, and multicollinearity. Given the use of listwise deletion, sample sizes varied across models based on the availability of complete data for independent, moderator, and dependent variables (*n* range: 139–149). All models controlled for participant age.

## 3. Results

Table 1 presents participant characteristics, including age (range: 18–67 years) and descriptive statistics for all key variables.

Pearson correlation coefficients are presented in Table 2. PTSD symptoms were significantly positively correlated with global sleep quality (r = 0.42, *p* < 0.0001) and all three types of discrimination (anticipated: r = 0.47; day-to-day: r = 0.47; major: r = 0.34; all *p* < 0.0001). The three discrimination subscales were strongly intercorrelated (r = 0.61–0.77, all *p* < 0.0001). Age was not significantly correlated with PTSD symptoms (r = −0.08, *p* = 0.34).

There were no significant interactions between anticipated discrimination and sleep measures with post-traumatic stress (Table 3).

There was one significant interaction between day-to-day discrimination and sleep measures (Table 4). The association between day-to-day discrimination and post-traumatic stress was significantly moderated by sleep quality (*p* = 0.0126), such that better sleep quality resulted in a stronger positive relationship between day-to-day discrimination and post-traumatic stress (see Figure 1).

Additionally, there was one significant interaction between major discrimination and sleep measures (Table 5). The association between major discrimination and post-traumatic stress was significantly moderated by sleep quality (*p* = 0.0235), such that better sleep quality resulted in a stronger positive relationship between major discrimination and post-traumatic stress (see Figure 2).

## 4. Discussion

This exploratory study examined the moderating role of sleep quality on relationships between multiple types of discrimination and the severity of PTSD symptoms among Black transgender women. Contrary to expectations based on research among general populations, better sleep quality strengthened rather than buffered positive associations between both day-to-day and major discrimination and PTSD symptom severity. These paradoxical findings demand cautious interpretation and acknowledgment of multiple plausible explanations, particularly given the exploratory nature of this secondary analysis and its methodological constraints.

### 4.1. Three Explanatory Frameworks

We propose three non-mutually exclusive explanations that may account for our unexpected results: a testable psychological hypothesis involving dissociation, systematic measurement validity issues, and structural determinants that fundamentally alter sleep’s relationship with mental health. Importantly, these frameworks are offered as tentative, theory-consistent possibilities rather than empirically demonstrated mechanisms in this specific population, and they require direct testing in future work.

#### 4.1.1. Framework 1: The Dissociation Hypothesis (Untested Speculation)

One possible explanation, though importantly, one we did not directly measure and therefore present as speculation requiring empirical testing, involves dissociation as an adaptive response to chronic trauma. Individuals experiencing poor sleep quality (characterized by hyperarousal, sleep fragmentation, and insomnia) may be more likely to employ dissociation as a psychological defense to cope with persistent discrimination-related distress [25]. Dissociation can manifest as emotional numbing or detachment from distressing experiences and has been documented in traumatized populations [26,27].

Those who sleep poorly may have reduced capacity to fully process or emotionally engage with traumatic events, leading to lower self-reported distress on standardized measures. Conversely, individuals with better sleep may have fewer dissociative symptoms, resulting in increased psychological awareness and emotional engagement with discrimination experiences, thereby reporting higher distress levels when exposed to discrimination. Research documents that up to 30% of gender diverse individuals have lifetime dissociative disorder diagnoses [28], and sleep normalization has been associated with reductions in dissociative symptoms [29].

However, several critical caveats must accompany this interpretation. First, we did not measure dissociation in this study; this remains a testable hypothesis for future research rather than an empirically supported explanation. Second, while dissociation may reduce acute self-reported distress, characterizing it as simply “protective” oversimplifies a complex phenomenon with high long-term costs. Chronic dissociation is associated with impaired day-to-day functioning, increased vulnerability to revictimization, interference with trauma processing and recovery, and difficulties maintaining interpersonal relationships [30]. Thus, if dissociation explains our findings, it represents an understandable adaptive response to overwhelming circumstances that carries substantial risks alongside any short-term defensive benefits. Third, the relationship between poor sleep and dissociation may itself reflect structural violence; chronic insomnia and hyperarousal are sequelae of living under constant threat in unsafe environments.

#### 4.1.2. Framework 2: The Measurement Validity Hypothesis and Cultural Appropriateness

A second, and in our view, equally plausible explanation is that these paradoxical findings reflect systematic measurement error rather than genuine psychological phenomena. The assessment instruments employed, the Pittsburgh Sleep Quality Index (PSQI) [22] and PTSD Symptom Checklist (PCL-5) [23,24], were developed and validated primarily among White, cisgender populations and may exhibit significant measurement nonequivalence for Black transgender women [31,32]. Although internal consistency for both measures ranged from acceptable to excellent in this sample (PSQI α = 0.70; PCL-5 α = 0.96), internal consistency alone does not establish cultural validity or measurement invariance.

Psychological assessment tools frequently exhibit measurement problems across cultural and demographic groups that only become apparent through comprehensive validity assessment [33]. While 88% of commonly used psychological measures appear valid when assessed using only internal consistency, only 4% demonstrate good validity when comprehensively evaluated for test-retest reliability, factor structure, and measurement invariance. This pattern illustrates that strong internal consistency can coexist with poor structural, factorial, or cross-cultural validity; measures may be internally consistent while still failing to capture the same constructs or item meanings across different populations.

Specific concerns with the PSQI for Black transgender women include the following: First, items asking about “overall sleep quality” may be interpreted differently by individuals experiencing housing instability, sleeping in unsafe environments, or sharing beds due to economic necessity; circumstances disproportionately affecting Black transgender women [34]. What researchers code as “good sleep quality” may not reflect genuinely restorative sleep when occurring in contexts of ongoing threat. Second, questions about “sleeping alone” or “having a comfortable sleeping environment” may not capture relevant dimensions of sleep quality for individuals navigating housing insecurity, residential segregation, or intimate partner violence. Third, the measure may inadequately capture sleep fragmentation resulting from chronic hypervigilance, a necessary survival response in unsafe neighborhoods or abusive housing situations.

Similarly, the PCL-5 presents cultural validity concerns. First, PTSD assessment tools show differential item functioning across racial and ethnic groups, with African American women showing different symptom presentations than White women [35,36]. Second, items about “feeling distant from others” may have fundamentally different meanings in contexts of chosen families, community marginalization, and necessary boundary-setting in response to discrimination. What clinicians interpret as “emotional numbing” may reflect adaptive selectivity about whom to trust. Third, questions about “difficulty experiencing positive emotions” may not account for culturally specific expressions of resilience, joy, and community connection that exist alongside trauma. Fourth, current PTSD frameworks inadequately capture ongoing, chronic discrimination as a continuous traumatic stressor distinct from discrete traumatic events [32].

The possibility that measurement bias explains our findings highlights a critical gap: the urgent need for culturally sensitive, intersectionally informed assessment approaches specifically validated for populations experiencing multiple, intersecting forms of marginalization [31,37].

#### 4.1.3. Framework 3: Structural Determinants and Material Conditions

A third explanatory framework, largely absent from the existing psychological literature, centers on structural violence and material conditions as fundamental determinants of the observed relationships. This framework posits that Black transgender women with “good sleep” according to standardized measures may nonetheless be sleeping in profoundly adverse circumstances that undermine sleep’s restorative and protective functions.

Black transgender women face extraordinarily high rates of housing instability, homelessness, economic marginalization, and residential segregation in high-poverty, high-crime neighborhoods with substantial environmental hazards [34]. Thus, individuals scoring well on sleep duration and continuity may still be sleeping in overcrowded, unsafe, or unstable housing; experiencing ongoing threat requiring hypervigilance even during sleep; living in neighborhoods with high ambient noise, poor air quality, and exposure to violence; lacking resources for comfortable sleeping environments, adequate heating or cooling, or privacy; sharing beds due to economic necessity or housing instability; experiencing food insecurity affecting sleep physiology; and managing chronic health conditions without adequate healthcare access.

Under these conditions, achieving “good sleep quality” as measured by duration and continuity may require enormous psychological resources and may not provide the same restorative benefits observed in privileged populations sleeping in safe, comfortable, resourced environments. The cognitive and emotional capacity required to process discrimination-related distress may be absent even when sleep duration appears adequate, because the sleep itself occurs under conditions of ongoing threat and deprivation.

Moreover, economic marginalization creates difficult choices: working multiple jobs to afford housing may require sleep at non-optimal circadian times; lack of transportation may necessitate long commutes, reducing sleep opportunity; survival sex work may require nighttime waking; and caring responsibilities for chosen family may fragment sleep. In this context, the positive correlation between “good sleep” and distress may reflect that those able to maintain sleep despite overwhelming circumstances are simultaneously most exposed to discrimination in employment, housing, and public spaces where they are visibly living their lives.

This structural determinant framework suggests that our findings do not reflect individual psychological processes but rather the inadequacy of decontextualized sleep and mental health measures to capture the lived reality of surviving while Black and transgender in the United States. The measurements may be detecting signals about structural conditions rather than, or in addition to, individual psychological states.

### 4.2. Critical Considerations and Future Research

Given current evidence, all three explanatory frameworks warrant serious consideration, and they are not mutually exclusive. Future research must urgently test the dissociation hypothesis by directly measuring dissociative symptoms alongside sleep quality, discrimination exposure, and PTSD symptoms using longitudinal designs. However, such research must first establish that the study measures themselves demonstrate cultural validity for Black transgender women.

Simultaneously, researchers must prioritize establishing cultural validity and measurement equivalence of assessment instruments through community-engaged processes. This includes extensive qualitative research to understand how Black transgender women conceptualize sleep quality, mental health, and coping; cognitive interviewing to identify how existing items are interpreted; comprehensive psychometric evaluation, including test-retest reliability, factor structure, and measurement invariance; and development of new or adapted measures informed by community input [31,38,39].

Most critically, future research must comprehensively assess structural determinants, including housing stability, quality, and safety; economic resources, including income, wealth, employment stability, and food security; neighborhood characteristics, including segregation, environmental quality, safety, and access to resources; healthcare access, including insurance status, regular care, discrimination experiences, and trust; experiences of violence and ongoing threat; and transportation access and commute burden. Only by measuring these structural factors can we determine whether they account for paradoxical sleep-distress relationships.

Longitudinal studies utilizing life course frameworks are essential to understand how cumulative exposure to structural racism, housing instability, economic marginalization, and discrimination shapes trajectories of sleep, dissociation, and mental health over time. Experience sampling or daily diary methods could clarify temporal dynamics while minimizing retrospective biases. The mixed method approaches, combining standardized measures with qualitative interviews, can identify when assessments miss important cultural phenomena.

### 4.3. Clinical Implications

Clinicians working with Black transgender women must approach these findings with substantial caution. We emphatically caution against misinterpreting these results as suggesting that poor sleep is beneficial or that sleep interventions are harmful for Black transgender women. Such an interpretation would be profoundly misguided and potentially dangerous. Given the modest sample size, lack of a control group, and use of measures not yet validated for this population, any clinical implications drawn from these findings should be regarded as preliminary and hypothesis-generating rather than practice-changing.

Within these constraints, the present study may still offer tentative implications for assessment and future intervention development, rather than direct guidance for clinical practice. Specifically, our findings highlight that standard assessment tools may not accurately capture mental health and sleep among multiply marginalized populations; improving sleep quality likely remains beneficial but may be insufficient without addressing structural determinants; interventions must be trauma-informed, culturally grounded, and attentive to structural barriers; and clinical decision-making may be better informed by comprehensive assessment including structural factors, not standardized scores alone.

If dissociation proves relevant, integrating assessment and support for dissociative symptoms alongside sleep and PTSD interventions may be important, while recognizing dissociation as an understandable response carrying both adaptive elements and high long-term costs. However, focusing solely on individual psychological interventions without addressing housing instability, economic vulnerability, discrimination, and violence will have limited effectiveness.

### 4.4. Strengths and Limitations

This study addresses significant research gaps by focusing on Black transgender women, a multiply marginalized population experiencing profound disparities. The use of the Intersectional Discrimination Index enables a nuanced examination of different discrimination types, and the sample size is substantial for this hard-to-reach population. The community-engaged approach, reflected in our Community Advisory Board partnership, strengthens cultural relevance and centers affected community expertise.

However, several important limitations warrant acknowledgment. The cross-sectional design precludes causal inference and temporal assessment. In addition, this study did not include a comparison or control group (e.g., cisgender women or transgender people of other racial backgrounds), so it is not possible to assess whether the observed patterns are unique to Black transgender women or reflect broader mechanisms operating across groups. We cannot determine whether sleep patterns influence PTSD symptoms, whether PTSD affects sleep, whether both are jointly influenced by unmeasured factors, or whether structural conditions create spurious associations.

Sample size limitations, while substantial for transgender research with objective measurement, constrain power for complex moderation and mediation analyses and may limit generalizability. Participants were recruited through community organizations and online platforms, potentially underrepresenting individuals who are most socially isolated, unstably housed, or lacking internet access; precisely those experiencing the greatest structural marginalization.

Most critically, we lacked comprehensive measures of numerous factors that theory suggests may be key to understanding our findings. We did not assess dissociation or dissociative symptoms, despite proposing this as a primary explanatory hypothesis; housing stability, quality, safety, or homelessness experiences; economic resources beyond basic demographics, including income instability, wealth, debt, and food security; neighborhood characteristics, including segregation indices, environmental quality, safety, noise levels, or access to resources; healthcare access quality, including insurance continuity, discrimination experiences in healthcare, or trust in medical institutions; experiences of ongoing violence, threats, or chronic danger; social support, chosen family, or community connection; previous trauma history beyond current PTSD symptoms; substance use or other coping mechanisms; physical health conditions or chronic pain; and employment stability, work schedules, or multiple job demands. Taken together, these limitations mean that the mechanisms we propose remain speculative.

Additionally, the use of binary racial categorization, while relevant for capturing structural racism experiences in the United States, oversimplifies heterogeneous experiences within racial groups. The study included only Black transgender women, limiting generalizability to other racial/ethnic groups or transgender masculine individuals.

The assessment instruments employed, PSQI and PCL-5, lack established cultural validity for Black transgender women, as discussed extensively above. This represents not merely a limitation but a fundamental threat to validity that may fully explain our paradoxical findings. The field urgently needs validated measures specifically designed for multiply marginalized populations.

## 5. Conclusions

This exploratory study reveals that better sleep quality strengthened rather than buffered associations between discrimination and PTSD symptoms among Black transgender women, a finding that contradicts patterns observed among general populations. We propose three plausible, non-mutually exclusive explanations: dissociation as an adaptive response (untested hypothesis requiring direct measurement), systematic measurement validity issues stemming from culturally inappropriate assessment tools, and structural determinants that fundamentally alter sleep’s relationship with mental health. However, because this was an exploratory secondary analysis without a control group and with notable measurement limitations, these mechanisms should be viewed as provisional and require confirmation in larger, comparative studies.

These exploratory findings suggest that psychological research developed in and validated for White, cisgender, socioeconomically advantaged populations cannot be uncritically applied to multiply marginalized communities. Paradoxical findings should prompt critical examination of measurement validity and structural context, not automatic pathologizing of minoritized populations.

The potential for measurement bias to produce misleading results when culturally inappropriate tools are applied to marginalized populations represents a critical threat to scientific validity and research justice [40]. Rather than accepting paradoxical findings as revealing psychological deficits, the field must prioritize developing and validating culturally sensitive, intersectionally informed measurement approaches. Simultaneously, research must comprehensively assess structural determinants that shape both sleep and mental health, recognizing that disparities reflect systemic oppression rather than individual pathology.

Future research integrating validated measures of dissociation, sleep, discrimination, trauma, and structural determinants is urgently needed. Only through such methodological rigor, combined with explicit attention to cultural validity and structural context, can researchers generate findings that truly serve the needs of Black transgender women and other multiply marginalized communities. Until validated instruments and comprehensive structural assessment become standard, clinical and policy recommendations based on paradoxical findings should be made with extreme caution and explicit acknowledgment of measurement and structural limitations.

Most fundamentally, these findings challenge psychology to move beyond locating health disparities within individual bodies and behaviors and instead recognize that divergent patterns reflect the embodiment of social inequities and inadequacy of decontextualized measurement. The field must commit to developing measurement approaches and analytical frameworks grounded in the lived realities of multiply marginalized communities, designed in partnership with those communities, and attentive to how structural violence shapes both health outcomes and their measurement.

## Figures and Tables

**Figure 1 healthcare-14-00137-f001:**
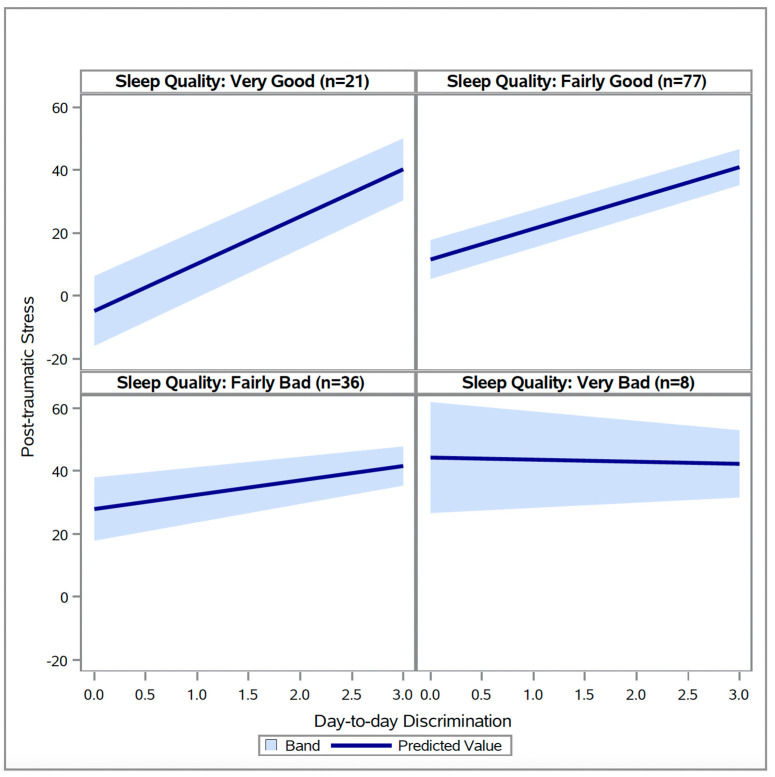
Relationships between day-to-day discrimination at different levels of sleep quality (N = 142).

**Figure 2 healthcare-14-00137-f002:**
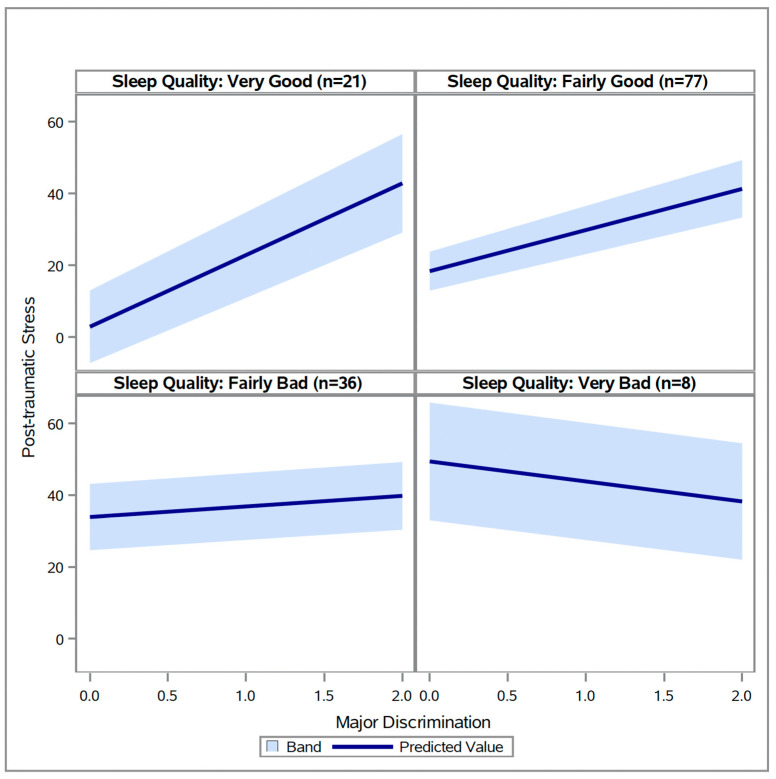
Relationships between major discrimination at different levels of sleep quality (N = 142).

**Table 1 healthcare-14-00137-t001:** Descriptive statistics.

	N	Mean	Std Dev
Age	155	36.13	11.01
Anticipated Discrimination	152	2.20	1.12
Day-to-day Discrimination	152	1.68	0.90
Major Discrimination	152	0.80	0.54
Post-Traumatic Stress	151	29.17	20.12
Sleep Quality	146	1.23	0.77
Sleep Latency	145	1.77	0.96
Sleep Duration	154	0.87	0.93
Sleep Efficiency	153	1.03	1.15
Sleep Disturbance	145	1.52	0.79
Sleep Medication Use	146	1.08	1.27
Daytime Dysfunction	146	1.18	0.88
Global PSQI Score	143	8.65	4.10

**Table 2 healthcare-14-00137-t002:** Pearson correlation coefficients.

	1	2	3	4	5	6	7	8	9	10	11	12
1. Global PSQI Score	-											
2. Sleep Quality	0.70 ***	-										
3. Sleep Latency	0.71 ***	0.54 ***	-									
4. Sleep Duration	0.63 ***	0.38 ***	0.41 ***	-								
5. Sleep Efficiency	0.51 ***	0.16 *	0.17 *	0.39 ***	-							
6. Sleep Disturbance	0.71 ***	0.54 ***	0.51 ***	0.32 ***	0.15	-						
7. Sleep Medication Use	0.56 ***	0.24 ***	0.28 ***	0.08	0.16	0.23 ***	-					
8. Daytime Dysfunction	0.49 ***	0.37 ***	0.18 *	0.13	−0.04	0.52 ***	0.16	-				
9. Post-Traumatic Stress	0.42 ***	0.34 ***	0.34 ***	0.18 *	0.014	0.47 ***	0.11	0.51 ***	-			
10. Anticipated Discrimination	0.22 **	0.22 ***	0.11	0.15	−0.03	0.27 ***	−0.02	0.35 ***	0.47 ***	-		
11. Major Discrimination	0.17 *	0.18 *	0.04	0.18 *	0.00	0.28 ***	−0.06	0.24 **	0.34 ***	0.61 ***	-	
12. Day-to-day Discrimination	0.17 *	0.21 *	0.02	0.14	−0.11	0.27 **	0.01	0.35 ***	0.47 ***	0.77 ***	0.71 ***	-
13. Age	0.14	0.05	0.01	0.18*	0.07	0.02	0.29 ***	−0.16	−0.08	−0.24 **	−0.12	−0.17 *

Prob > |r| under H0: Rho = 0; * *p* < 0.05, ** *p* < 0.01, *** *p* < 0.001; two-tailed.

**Table 3 healthcare-14-00137-t003:** Associations between anticipated discrimination x sleep and post-traumatic stress.

Parameter	b	SE	*p*	Model Statistics
Intercept	−2.32	7.63	0.7617	F(4,137) = 13.22, *p* < 0.001 R^2^ = 0.279 (0.257) N = 142
Age	0.06	0.14	0.6721	
Anticipated Discrimination	9.94	2.33	<0.0001 ***	
Sleep Quality	10.65	4.24	0.0132 *	
Anticipated Discrimination × Sleep Quality	−2.07	1.65	0.2117	
Intercept	−5.48	8.18	0.5040	F(4,136) = 16.03, *p* < 0.001 R^2^ = 0.320 (0.300) N = 141
Age	0.06	0.13	0.6729	
Anticipated Discrimination	9.48	2.74	0.0007 ***	
Sleep Latency	7.81	3.08	0.0123 *	
Anticipated Discrimination × Sleep Latency	−0.70	1.31	0.5916	
Intercept	6.67	6.83	0.3303	F(4,144) = 11.58, *p* < 0.001 R^2^ = 0.243 (0.222) N = 149
Age	0.01	0.14	0.9622	
Anticipated Discrimination	9.15	1.75	<0.0001 ***	
Sleep Duration	4.52	3.55	0.2056	
Anticipated Discrimination × Sleep Duration	−0.97	1.38	0.4831	
Intercept	10.19	7.00	0.1476	F(4,143) = 11.14, *p* < 0.001 R^2^ = 0.238 (0.216) N = 148
Age	0.04	0.14	0.7533	
Anticipated Discrimination	7.49	1.79	<0.0001 ***	
Sleep Efficiency	−2.15	2.74	0.4328	
Anticipated Discrimination × Sleep Efficiency	1.29	1.16	0.2688	
Intercept	−3.89	7.87	0.6221	F(4,136) = 18.22, *p* < 0.001 R^2^ = 0.349 (0.330) N = 141
Age	0.04	0.13	0.7845	
Anticipated Discrimination	7.74	2.60	0.0034 *	
Sleep Disturbance	10.98	3.85	0.0050 *	
Anticipated Discrimination × Sleep Disturbance	−0.71	1.56	0.6496	
Intercept	6.24	7.09	0.3801	F(4,137) = 10.35, *p* < 0.001 R^2^ = 0.232 (0.210) N = 142
Age	0.04	0.15	0.8113	
Anticipated Discrimination	8.84	1.78	<0.0001 ***	
Sleep Medication Use	2.02	2.79	0.4700	
Anticipated Discrimination × Sleep Medication Use	−0.21	1.12	0.8531	
Intercept	−0.56	6.93	0.9353	F(4,137) = 18.91, *p* < 0.001 R^2^ = 0.356 (0.337) N = 142
Age	0.18	0.13	0.1686	
Anticipated Discrimination	5.24	2.11	0.0141 *	
Daytime Dysfunction	7.38	3.80	0.0542	
Anticipated Discrimination × Daytime Dysfunction	0.81	1.57	0.6084	
Intercept	−1.66	8.00	0.8359	F(4,134) = 16.13, *p* < 0.001 R^2^ = 0.325 (0.305) N = 139
Age	−0.02	0.14	0.8727	
Anticipated Discrimination	7.75	2.72	0.0050 **	
Global PSQI Score	1.78	0.72	0.0150 *	
Anticipated Discrimination × Global PSQI Score	−0.08	0.29	0.7752	

Abbreviations: b = unstandardized beta; SE = standard error; * *p* < 0.05, ** *p* < 0.01, *** *p* < 0.001; two-tailed. Note. Model statistics are presented as follows: the first line reports the overall model F statistic with numerator and denominator degrees of freedom and associated *p*-value; the second line reports R^2^ with adjusted R^2^ in parentheses; and the third line reports the analytic sample size (N).

**Table 4 healthcare-14-00137-t004:** Associations between day-to-day discrimination x sleep and post-traumatic stress.

Parameter	b	SE	*p*	Model Statistics
Intercept	−4.54	7.62	0.5521	F(4,137) = 13.70, *p* < 0.001 R^2^ = 0.286 (0.265) N = 142
Age	−0.01	0.14	0.9555	
Day-To-Day Discrimination	15.00	3.02	<0.0001 ***	
Sleep Quality	16.35	4.38	0.0003 ***	
Day-To-Day Discrimination × Sleep Quality	−5.22	2.07	0.0126 *	
Intercept	−3.58	8.46	0.6723	F(4,136) = 15.62, *p* < 0.001 R^2^ = 0.315 (0.295) N = 141
Age	−0.01	0.13	0.9388	
Day-To-Day Discrimination	11.76	3.78	0.0023	
Sleep Latency	8.99	3.45	0.0102 *	
Day-To-Day Discrimination × Sleep Latency	−1.04	1.84	0.5705	
Intercept	10.68	6.56	0.1058	F(4,144) = 10.97, *p* < 0.001 R^2^ = 0.233 (0.212) N = 149
Age	−0.07	0.14	0.6366	
Day-To-Day Discrimination	10.97	2.26	<0.0001 ***	
Sleep Duration	4.65	3.29	0.1607	
Day-To-Day Discrimination × Sleep Duration	−1.15	1.65	0.4856	
Intercept	14.74	6.83	0.0325	F(4,143) = 10.95, *p* < 0.001 R^2^ = 0.234 (0.213) N = 148
Age	−0.03	0.14	0.8470	
Day-To-Day Discrimination	8.16	2.33	0.0006 ***	
Sleep Efficiency	−1.94	2.53	0.4452	
Day-To-Day Discrimination × Sleep Efficiency	2.07	1.40	0.1430	
Intercept	−2.62	7.93	0.7417	F(4,136) = 17.05, *p* < 0.001 R^2^ = 0.334 (0.314) N = 141
Age	−0.02	0.13	0.8692	
Day-To-Day Discrimination	10.57	3.71	0.0051 **	
Sleep Disturbance	12.78	3.84	0.0011 **	
Day-To-Day Discrimination × Sleep Disturbance	−1.85	2.04	0.3667	
Intercept	12.28	7.03	0.0830	F(4,137) = 8.99, *p* < 0.001 R^2^ = 0.208 (0.185) N = 142
Age	−0.04	0.15	0.7986	
Day-To-Day Discrimination	9.50	2.29	<0.0001 ***	
Sleep Medication Use	1.16	2.59	0.6547	
Day-To-Day Discrimination × Sleep Medication Use	0.38	1.36	0.7805	
Intercept	1.56	6.84	0.8199	F(4,137) = 17.40, *p* < 0.001 R^2^ = 0.337 (0.318) N = 142
Age	0.14	0.13	0.2983	
Day-To-Day Discrimination	6.56	2.68	0.0157 *	
Daytime Dysfunction	9.31	3.24	0.0047 **	
Day-To-Day Discrimination × Daytime Dysfunction	0.07	1.68	0.9653	
Intercept	−1.09	8.20	0.8944	F(4,134) = 15.76, *p* < 0.001 R^2^ = 0.320 (0.300) N = 139
Age	−0.08	0.14	0.5349	
Day-To-Day Discrimination	10.36	3.70	0.0058 **	
Global PSQI Score	2.16	0.73	0.0035 **	
Day-To-Day Discrimination × Global PSQI Score	−0.23	0.37	0.5287	

Abbreviations: b = unstandardized beta; SE = standard error; * *p* < 0.05, ** *p* < 0.01, *** *p* < 0.001; two-tailed. Note: Model statistics are presented as follows: The first line reports the overall model F statistic with numerator and denominator degrees of freedom and associated *p*-value. The second line reports R^2^ with adjusted R^2^ in parentheses and the third line reports the analytic sample size (N).

**Table 5 healthcare-14-00137-t005:** Associations between major discrimination x sleep and post-traumatic stress.

Parameter	b	SE	*p*	Model Statistics
Intercept	4.73	7.48	0.5284	F(4,137) = 9.20, *p* < 0.001 R^2^ = 0.212 (0.189) N = 142
Age	−0.05	0.14	0.7105	
Major Discrimination	20.04	5.30	0.0002 ***	
Sleep Quality	15.53	4.10	0.0002 ***	
Major Discrimination × Sleep Quality	−8.55	3.73	0.0235 *	
Intercept	7.78	7.78	0.3192	F(4,136) = 9.66, *p* < 0.001 R^2^ = 0.221 (0.198) N = 141
Age	−0.08	0.14	0.5519	
Major Discrimination	13.82	6.39	0.0323 *	
Sleep Latency	8.15	2.99	0.0072 **	
Major Discrimination × Sleep Latency	−1.19	3.15	0.7051	
Intercept	19.05	6.34	0.0031 **	F(4,144) = 6.18, *p* < 0.001 R^2^ = 0.146 (0.123) N = 149
Age	−0.14	0.15	0.3439	
Major Discrimination	15.65	4.04	0.0002 ***	
Sleep Duration	7.39	3.27	0.0255 *	
Major Discrimination × Sleep Duration	−5.19	3.24	0.1118	
Intercept	21.64	6.52	0.0012 **	F(4,143) = 4.73, *p* = 0.001 R^2^ = 0.117 (0.092) N = 148
Age	−0.09	0.15	0.5576	
Major Discrimination	12.34	3.97	0.0023 **	
Sleep Efficiency	0.28	2.47	0.9109	
Major Discrimination × Sleep Efficiency	0.08	2.66	0.9754	
Intercept	5.42	7.44	0.4674	F(4,136) = 12.52, *p* < 0.001 R^2^ = 0.269 (0.248) N = 141
Age	−0.09	0.14	0.5251	
Major Discrimination	13.12	6.44	0.0437 *	
Sleep Disturbance	13.76	3.71	0.0003 ***	
Major Discrimination × Sleep Disturbance	−3.74	3.82	0.3295	
Intercept	20.55	6.45	0.0018 **	F(4,137) = 4.91, *p* < 0.001 R^2^ = 0.125 (0.100) N = 142
Age	−0.13	0.16	0.3992	
Major Discrimination	13.15	3.84	0.0008 ***	
Sleep Medication Use	2.90	2.34	0.2174	
Major Discrimination × Sleep Medication Use	−0.81	2.44	0.7407	
Intercept	6.28	6.76	0.3546	F(4,137) = 14.71, *p* < 0.001 R^2^ = 0.300 (0.280) N = 142
Age	0.11	0.14	0.4406	
Major Discrimination	7.47	5.00	0.1377	
Daytime Dysfunction	10.62	3.06	0.0007 ***	
Major Discrimination × Daytime Dysfunction	0.08	3.38	0.9804	
Intercept	3.64	7.60	0.6326	F(4,134) = 11.66, *p* < 0.001 R^2^ = 0.258 (0.236) N = 139
Age	−0.16	0.14	0.2445	
Major Discrimination	17.88	6.54	0.0071 **	
Global PSQI Score	2.75	0.69	0.0001 ***	
Major Discrimination × Global PSQI Score	−1.01	0.69	0.1427	

Abbreviations: b = unstandardized beta; SE = standard error; * *p* < 0.05, ** *p* < 0.01, *** *p* < 0.001; two-tailed. Note: Model statistics are presented as follows: The first line reports the overall model F statistic with numerator and denominator degrees of freedom and associated *p*-value. The second line reports R^2^ with adjusted R^2^ in parentheses and the third line reports the analytic sample size (N).

## Data Availability

The data presented in this study are available on request from the corresponding author. The data are not publicly available due to privacy restrictions.

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
