# Peer review of "Discrimination and Symptoms of Post-Traumatic Stress Among Black Transgender Women in the United States: The Moderating Effect of Sleep"

_healthcare, 2026, doi:10.3390/healthcare14020137_

Round 1

Reviewer 1 Report

Comments and Suggestions for Authors

Dear authors, thank you for the opportunity to review your work. I found it very interesting.

Upon reviewing the calculations performed to assess the moderating effect of the variables considered between the independent variables (experiences of anticipatory discrimination, experiences of daily discrimination, and experiences of major discrimination) and the dependent variable (post-traumatic stress), the identification and comparison of direct effects (relationship between independent and dependent variables without considering the influence of the moderators) and indirect effects (relationship between the variables using the moderator pathway) are particularly relevant. I believe this aspect is crucial for interpreting the moderating capacity of the variables considered.

A reference that may be helpful in clarifying this analytical procedure can be found (among others) in the work of Professor Rockwood: Advancing the Formulation and Testing of Multilevel Mediation and Moderated Mediation Models.

------

Title: Discrimination and Symptoms of Post-Traumatic Stress among Black Transgender Women in the United States: The Moderating Effect of Sleep

Author´s

Reviewer

Research objective:

This study examines how sleep quality moderates relationships between different  forms of discrimination (anticipated, day-to-day, and major) and PTSD symptom severity  among Black transgender women. We approach this inquiry with openness to the possibility that conventional protective effects of sleep quality may not operate as anticipated within this population.

This study examines how sleep quality moderates the relationships between different forms of discrimination (anticipated, daily, and major) and the severity of PTSD symptoms among Black transgender women. We approach this research with an openness to the possibility that the conventional protective effects of sleep quality may not operate as anticipated within this population.

Participants

Between 139 and 149    

Table 1 shows N values ​​between 143 and 155 (review text)

Variables:

a)      Independent variables:

a.1 Experiences of ANTICIPATED discrimination

a.2. Experiences of DAILY discrimination

               a.3. Experiences of MAJOR discrimination

b) Dependent variable:

Post-traumatic stress

c)      c) Moderating variables: Self-reported sleep components

1.      Quality

2.      Latency

3.      Duration

4.      Efficiency

5.      Alterationn

6.      Use of medications

7.      Overall score (Pittsburgh Sleep Quality Scale)

Statistical análisis

Separate multiple linear regression models

Table 2.

As an example:

I have interpreted the coefficient value of the model that does not take the moderator into account (Anticipated Discrimination  Sleep and Post-Traumatic Stress) as the one shown in the table for the Anticipated Discrimination variable.

I have not been able to identify the coefficient value that relates the independent variable to the moderator variable.

(see attached file)

In cases where the value of the interaction is not significant, there is no room to talk about moderation.

In the case of Table 2, there does not appear to be any moderation with any of the variables used.

In the cases of (a) Sleep Quality, (b) Sleep Latency, (c) Sleep Disturbance, (d) Daytime Dysfunction, and Global PSQI Score, their relationship with Sleep and Post-Traumatic Stress was significant.

However, the relationship between Anticipated Discrimination and Sleep Quality does not appear to be significant. This explains why the interaction between the two is not significant.

There is significance in:

a)      Table 3: Day-to-day Discrimination * Sleep Quality

b)     Table 4: Major Discrimination * Sleep Quality

It is very likely that the relationship between:

Anticipated Discrimination (as an independent variable) and Sleep Quality (as a dependent variable) is not significant.

Discussion

It may be necessary to reinterpret the results obtained.

Reviewer 2 Report

Comments and Suggestions for Authors

This paper presents an interesting analysis of how sleep quality moderates the relationships between different forms of discrimination and the severity of post-traumatic stress symptom among black transgender women in the United States.

I believe that the topic is relevant and interesting. The paper is well written and easy to read. However, some improvements should be considered:

The abstract places the study in a broader context, contains a brief description of the main methods, outlines the main results, and provides a summary of the main conclusions, but does not adequately clarify the purpose of the study.

The introduction provides a broader context for the study, highlights its importance, and briefly mentions the main objective of the study in a manner that is easy to understand even for readers who are not familiar with the topic. However, I recommend providing a clearer perspective on the current state of research in the field, with a more consistent analysis of the main publications that provide a theoretical framework for this endeavour. 

The section on methodology is logically structured.

The results of the study demonstrate a consistent research effort. To improve the clarity of the tables and, by extension, of the results, I suggest adding more detailed legends to clarify the meaning of statistical abbreviations.

In the Discussions section, the results are correlated with relevant scientific literature and the limitations of the paper are well highlighted.

Reviewer 3 Report

Comments and Suggestions for Authors

Please find the comments:

  1. The abstract is too long. See the requirements.
  2. Add more keywords (up to ten) to improve indexation.
  3. Conclusions in the abstract are speculations which do not refer to specific results of this study. Reconsider.
  4. The introduction is too short and does not provide sufficient background for this study. I would suggest providing hypotheses with their justification based on the literature.
  5. Exclusion criteria should be mentioned.
  6. Measures should be described in more detail. Please include examples of items. 
  7. Internal consistency reliability of the measures used in this study should be calculated. 
  8. "Sample size differed slightly across models based on complete
    data from independent, moderator, and dependent variables (n range: 139-149). All
    models controlled for participant age." Where 139-149? In Table 1 it is from 143 to 155.
  9. Simple correlation between measures were not computed. Reconsider. 
  10. Min and max age were not indicated.
  11. Before Table 1, there is no text. Please prepare the paper according to high standards. In the current form, it looks not well, with basic methodological, statistical, and language flaws, including stylistic elements.
  12. Many models were tested. Please correct for multiple testing.
  13. The sample size is small to test multiple groups with different levels of sleep quality (n = 21, 77, 36 and 8). it is unclear on what the labels of good sleep quality, bad sleep quality etc. were based.
  14. The discussion refers to paradoxical results. There is no surprise as there methodological flaws which may impact the results. Overall, the discussion is very speculative.
  15. Implications of the study are also too speculation and irrelevant taking into account the small sample size and methodological flaws. The control group was not provided, and therefore potential differences between the examined group and the control group were not tested. As such, it is unclear whether mechanisms are different in these groups.

Overall, the study represents very preliminary results in a small sample, based on cross-sectional research, with very strong overinterpretations and methodological issues. 

Round 2

Reviewer 1 Report

Comments and Suggestions for Authors

The authors made most of the proposed changes. The rest they justified reasonably.

Reviewer 3 Report

Comments and Suggestions for Authors

1. Internal consistency reliability should be calculated for PSQI subscales too as they were used as moderator. Cronbach's alpha for the total PSQI score was calculated, but for subscales not.

2. Correlations should be presented with two decimal places. Only p-values should be presented with three decimal places.

3. Table 2: Usually, asterisks (*, **, ***) are used to indicate p-values rather than letters a, b, c, d etc.

4. Did you collect data on demographics except age? Or other significant clinical/subclinical variables which may impact the variables of interest. If yes, please present these.

5. Lines 80-82: Please elaborate on how these data were collected? Based on self-report? 

6. Regressions should be presented with basic statistical output (e.g., F-values, degrees of freedom, explained variance etc.). Please elaborate on fulfilling requirements for conducting regressions.
